# Claudin-1, A Double-Edged Sword in Cancer

**DOI:** 10.3390/ijms21020569

**Published:** 2020-01-15

**Authors:** Ajaz A. Bhat, Najeeb Syed, Lubna Therachiyil, Sabah Nisar, Sheema Hashem, Muzafar A. Macha, Santosh K. Yadav, Roopesh Krishnankutty, Shanmugakonar Muralitharan, Hamda Al-Naemi, Puneet Bagga, Ravinder Reddy, Punita Dhawan, Anthony Akobeng, Shahab Uddin, Michael P. Frenneaux, Wael El-Rifai, Mohammad Haris

**Affiliations:** 1Division of Translational Medicine, Research Branch, Sidra Medicine, Doha 26999, Qatar; abhat@sidra.org (A.A.B.); nsyed@sidra.org (N.S.); snisar1@sidra.org (S.N.); shashem@sidra.org (S.H.); syadav@sidra.org (S.K.Y.); 2Translational Research Institute, Academic Health System, Hamad Medical Corporation, Doha 3050, Qatar; LTherachiyil@hamad.qa (L.T.); RKrishnankutty@hamad.qa (R.K.); SKhan34@hamad.qa (S.U.); 3Department of Pharmaceutical Sciences, College of Pharmacy, QU Health, Qatar University, Doha 2713, Qatar; 4Department of Biotechnology, Central University of Kashmir, Ganderbal, Jammu and Kashmir 191201, India; muzafar.macha1@gmail.com; 5Department of Biochemistry and Molecular Biology, University of Nebraska Medical Center, Omaha, NE 68198, USA; punita.dhawan@unmc.edu; 6Laboratory Animal Research Center, Qatar University, Doha 2713, Qatar; smurli28@qu.edu.qa (S.M.); halnaemi@qu.edu.qa (H.A.-N.); 7Center for Magnetic Resonance and Optical Imaging, Department of Radiology, Perelman School of Medicine at the University of Pennsylvania, Philadelphia, PA 19104, USA; puneetb@pennmedicine.upenn.edu (P.B.); krr@pennmedicine.upenn.edu (R.R.); 8Department of Pediatric Gastroenterology, Sidra Medicine, Doha 26999, Qatar; aakobeng@sidra.org; 9Academic Health System, Hamad Medical Corporation, Doha 3050, Qatar; mfrenneaux@hamad.qa; 10Department of Surgery, University of Miami Miller School of Medicine, Miami, FL 33136, USA; welrifai@med.miami.edu

**Keywords:** claudin 1, tight junctions, tumor, metastasis, epithelial to mesenchymal transition

## Abstract

Claudins, a group of membrane proteins involved in the formation of tight junctions, are mainly found in endothelial or epithelial cells. These proteins have attracted much attention in recent years and have been implicated and studied in a multitude of diseases. Claudins not only regulate paracellular transepithelial/transendothelial transport but are also critical for cell growth and differentiation. Not only tissue-specific but the differential expression in malignant tumors is also the focus of claudin-related research. In addition to up- or down-regulation, claudin proteins also undergo delocalization, which plays a vital role in tumor invasion and aggressiveness. Claudin (CLDN)-1 is the most-studied claudin in cancers and to date, its role as either a tumor promoter or suppressor (or both) is not established. In some cancers, lower expression of CLDN-1 is shown to be associated with cancer progression and invasion, while in others, loss of CLDN-1 improves the patient survival. Another topic of discussion regarding the significance of CLDN-1 is its localization (nuclear or cytoplasmic vs perijunctional) in diseased states. This article reviews the evidence regarding CLDN-1 in cancers either as a tumor promoter or suppressor from the literature and we also review the literature regarding the pattern of CLDN-1 distribution in different cancers, focusing on whether this localization is associated with tumor aggressiveness. Furthermore, we utilized expression data from The Cancer Genome Atlas (TCGA) to investigate the association between CLDN-1 expression and overall survival (OS) in different cancer types. We also used TCGA data to compare CLDN-1 expression in normal and tumor tissues. Additionally, a pathway interaction analysis was performed to investigate the interaction of CLDN-1 with other proteins and as a future therapeutic target.

## 1. Introduction

Claudins and occludin, a group of cell junctional proteins, serve as the backbone of the tight junctions. Claudin family members perform dual roles; some have barrier activities, while others mediate the permeability of small molecules and ions. In addition to the localization pattern, the differential expression of claudins between normal and tumor tissue has drawn attention to these proteins as potential prime candidates for future cancer therapy. Another hot topic of discussion is the tumor-promoter or tumor-suppressor role of claudins. This opens a wide area of research in elucidating how the tissue-specific expression of claudins and their interaction with other molecules in the cell may result in these two opposing effects. Both defective tight junctions and the absence of tight junctions have shown to be associated with the development and progression of certain cancers. In this article, we include a brief introduction of tight junctions, the structure of claudins, and their role in various cancers. We also perform the bioinformatics analysis on TCGA data to supplement the literature review.

### 1.1. Tight Junctions

Adjacent epithelial cells are sealed into an epithelial barrier by the most apical intercellular junctions called tight junctions. Tight junctions, as a network of continuous strands, separate the plasma membrane into apical and basolateral domains [1,2]. Tight junctions between adjacent cells associate to form paired strands imparting mechanical strength to the cells [3,4] and serve as barriers to control the movement of small molecules and ions across the paracellular space [5,6,7]. Apart from their mechanical strength, maintaining polarity and paracellular movement, tight junction proteins are able to recruit signaling proteins for various cellular processes [4]. Alterations in the structure and function of tight junctions result in a multitude of diseases, especially adenocarcinoma of various organs [8,9,10]. The failure of tight junctiontight junctions or tight junction proteins is one of the many key factors that contribute to the progression of cancer, but this is not a universal phenomenon as there can be many other direct or indirect factors that contribute to the development of cancer. The second reason that loss of tight junctions or tight junction proteins is not a universal phenomenon to the development of cancers is that besides the epithelial cancers, there are also non-epithelial tumors such as small subset of laryngeal neoplasms [11], angiomas, lipomas and neuromas [12] which do not display failure of tight junctions but other contributing factors come into play.

### 1.2. Claudins

Claudins are integral to the structure and function of tight junctions with four membrane-spanning regions, which include two extracellular loops, N- and C-terminal cytoplasmic domains. The extracellular loops are highly conserved, and the C-terminal domain is important for localizing the claudins to tight junctions (Figure 1). Being a part of a multigene family, there are about 27 members of claudins that are unique in their tissue-specific expression and their molecular weight ranges from 20–34 kDa [3]. Claudins play an important role in regulating transepithelial permeability by regulating the epithelium’s paracellular permeability to small molecules and ions [5,7,13]. Post-translational modifications such as phosphorylation alter the paracellular functions of claudins, which in turn modulate diverse signal mechanisms [14,15,16,17]. 

### 1.3. Claudins and Cancer

One of the important factors in cellular transformation and tumorigenesis is the loss of cell-to-cell adhesion [1]. Accordingly, the claudin family of proteins is significantly involved in the progression and growth of several cancers [9,10,18]. Tumor progression is characterized by migration, invasion, and metastasis of cancer cells. Claudins are believed to play a significant role in these processes as their loss contributes to the loss of cell junctions in a tissue-dependent manner [18,19]. Claudins have been also reported to play a vital role in the epithelial–mesenchymal transition (EMT) (Figure 2), a process that favors the spread of carcinomas, generation of cancer stem cells (CSCs) or tumor-initiating cells (TICs), and chemo-resistance [20,21,22,23]. The loss of claudins in epithelial cells results in disrupted tight junction function responsible for impaired cell polarity and epithelial integrity [6,7]. Several studies have reported the mislocalization and altered expression of claudins in various cancers [19,24]. The CLDN-1 and CLDN-7 members of the claudin family are primarily found to be downregulated in several invasive cancers including breast, esophageal, and prostate cancers [9,19,25,26,27]. However, in contrast, overexpression of CLDN-1 has been observed in colon, nasopharyngeal, ovarian and oral squamous cell cancers [9,10], while CLDN-3 and -4 are highly overexpressed in ovarian cancer and upregulated in breast, gastric, pancreatic, prostate and uterine cancers [28,29,30]. Human carcinomas such as those of the breast, liver, ovary, prostate, colon, liver and stomach are found to exhibit altered expression of claudins [19]. The expression and localization patterns of some of the claudins serve as an important prognostic predictor in many cancers [30,31]. The consensus of whether claudin expression increases or decreases during tumorigenesis is still a debatable topic and open to more research. 

## 2. Claudin-1 and Cancer; Tumor Promoter or Suppressor

CLDN-1 is a membrane protein that, along with occludin and other claudins form the backbone of the tight junctions and is essential for epithelial barrier functions [32]. It was the first member of the claudin family to be identified with a molecular weight of 22 kDa and is strongly expressed in the intestine, spleen, brain, liver, kidney, and testis [19,33]. Studies have shown the direct involvement of CLDN-1 in the development and progression of several cancers, such as colon cancers [34], oral squamous cell carcinomas [35], breast cancers [36], melanomas [37,38], and in many other cancers as discussed in this review. In some cancers, CLDN-1 has the opposite role where the decreased expression of CLDN-1 is associated with cancer progression, invasion and development of the metastatic phenotype [37,39]. The expression of CLDN-1 in different types of cancer and cancer subtypes is summarized in Table 1. Based on the literature, CLDN-1 is one of the most deregulated claudins in human cancer and can function as a tumor promoter or suppressor depending on the type of cancer (Figure 3) (Table 2). The role of CLDN-1 as a tumor promoter is mostly through its effect on the invasion or motility of cancer cells. Considering the importance of claudins in cancer, targeting claudin expression appears to have promise in the treatment of cancer. The specific role of CLDN-1 in various cancers is discussed in the following sections.

### 2.1. Claudin-1 and Breast Cancer

Breast cancer is the second major cause of death in women, and its heterogeneous molecular nature is a significant obstacle in treatment planning [65]. It has several subtypes, such as human epidermal growth factor receptor 2 (HER2), triple-negative or basal-like, Luminal A and Luminal B type depending on the presence or absence of several hormone receptors like HER2, estrogen, and progesterone [66,67,68]. Recently, another subtype of breast cancer known as the claudin-low subtype has been reported [67]. Each subtype of breast cancer exhibits unique prognostic features and different molecular markers [69]. 

The *CLDN-1* gene has been found to be upregulated during the early involution of the mammary gland [70]. The differential expression of CLDN-1 observed in different cancers outlines the complexity of the potential role that it plays in the cancer process. The CLDN-1 expression level in breast cancer differs depending on the cancer subtypes [71]. Studies have shown a correlation between increased malignancy, invasiveness and recurrence of breast cancer with total or partial loss of CLDN-1 expression [36,70]. In most of the invasive human breast cancers such as ER+ luminal A and luminal B, CLDN-1 expression is found to be downregulated, while an increased expression and cytoplasmic delocalization of CLDN-1 has been observed in some of the aggressive ER- basal-like breast cancer (BLBC) subtypes [40,72,73]. CLDN-1 is also found to be downregulated in HER2 enriched and claudin low breast cancer subtypes [41]. CLDN-1 acts as a tumor suppressor in ER+ and as a tumor promoter in ER- cancer subtypes [25]. In hereditary and sporadic breast cancer, CLDN-1 is found to be involved in tumorigenesis by suppressing the proliferation of mammary epithelial cells [74]. Further, CLDN-1 overexpression in MDA-MB 361 breast cancer cells resulted in increased apoptosis [75,76]. While one study reported that the activation of CLDN-1 was repressed by the binding of E-cadherin to CLDN-1 promoter [77], knockdown of CLDN-1 has been found to be associated with decreased cell migration and induction of EMT in breast cancer cells [76]. Another study showed a unique pattern of expression for CLDN-1 in ER-ve and ER+ve tumors. The authors showed that the protein expressions of CLDN-1 were significantly higher in the basal-like subtype of breast cancers (ER-ve, Her-2-ve, EGFR+ve, CK5/6+ve, a subtype largely linked to poor outcome [40]. CLDN-1 expression has also been observed in a small percentage of invasive human breast cancers that exhibit different pathological lesions leading to complexity in CLDN-1 expression [78]. CLDN-1 also possesses tumor-promoting effects by increasing cell migration and by exhibiting anti-apoptotic effects in some breast cancer cell lines like MCF-7 [76,79]. 

Several proteins interact with CLDN-1 to fuel the progression of breast cancer, including the following: Ephrin B1, ESCRT, CD9 and EpCAM [80,81,82,83]. CLDN-1 mediates the tyrosine phosphorylation of Ephrin B1, a transmembrane protein, in a receptor independent manner which provides the evidence that ephrin-B1 inhibits the formation of the tight cell–cell adhesion in a wide variety of epithelial and cancer cells regardless of the existence of cognate Eph receptors [80]. Endosomal sorting complexes required for transport (ESCRT) machinery are a set of proteins present in the cytosol that are involved in the maintenance of cell polarity and the regulation of membrane-bound proteins [81]. When the function of ESCRT is inhibited, CLDN-1 accumulates in the cytoplasm causing the tight junctions to disassemble and lose cell polarity [25]. The loss of ESCRT function is also linked with increased proliferation and less stable tissue structure in the cancer cells. CLDN-1 is also found to interact with CD9, a transmembrane protein that regulates cell migration, proliferation, differentiation and fusion [82]. CD9 prevents the association between CLDN-1 and tight junctions that could cause the progression of the tumor. The subcellular co-localization of CLDN-1 and CD9 supports their interaction, and this was confirmed in many cell clines including different human breast cancer cell lines [82]. EpCAM (also known as epithelial cell adhesion molecule), another surface transmembrane glycoprotein known to be expressed in some invasive carcinomas is involved in cell proliferation and metastasis and has been shown to protect CLDN-1 from degradation. [83]. This could be a cause for the cytoplasmic accumulation of CLDN-1 in some breast cancer cell lines [76,83]. Several transcript variants for CLDN-1 were found in human invasive breast cancer as a result of splicing and mis splicing events suggesting that through alternative splicing CLDN-1 is downregulated in invasive type of breast cancers [72].

### 2.2. Claudin-1 and Thyroid Cancer

Thyroid cancer is the most commonly occurring endocrine malignancy [84,85]. A study by Nemeth et al. performed independent microarray expression analyses of two types of thyroid carcinomas, namely papillary thyroid cancer (PTC) and follicular thyroid cancer (FTC) [42]. The study showed that high expression of CLDN-1 is specific for the regional lymph node metastasis associated with PTC [42] and found increased expression of *CLDN-1* gene in PTC [86,87]. Sobel et al. reported high levels of CLDN-1 expression in serous papillary endometrial carcinoma [88]. Another study about the role of CLDN-1 in follicular-cell derived thyroid carcinoma cell lines (FTC-133 and FTC-238) found higher expression of CLDN-1 in the nuclei of FTC-238 cells as compared to the FTC-133 cells [43]. The same study demonstrated the increased pathogenic character of FTC-133 cells by RASV12 transfection was associated with high expression of CLDN-1 and enhanced cell proliferation and migration [43]. Conversely, the downregulation of CLDN-1 by siRNA caused decreased cell invasion and migration accompanied by decreased phospho-PKC expression in the FTC-238 cells, suggesting that the aggressiveness of follicular thyroid carcinoma associated with high CLDN-1 expression can be influenced by PKC activity [43]. Another study described the reduced expression of CLDN-1 in follicular carcinomas vs adenomas, specifically in the poorly-differentiated and undifferentiated types of human thyroid carcinomas [89]. The expression of CLDN-1 was significantly different between malignant and benign thyroid neoplasms, and between follicular and papillary carcinomas [90]. Similarly, papillary carcinomas showed significantly higher positive CLDN-1 expression. While negative CLDN-1 expression was observed in the tissue samples of normal thyroid and solitary-follicular-patterned-nodules [91].

### 2.3. Claudin-1 and Colorectal Cancer 

Colorectal cancer (CRC) is the fourth leading cause of cancer-related deaths and the third most frequently diagnosed malignancy worldwide [92]. Increased expression of CLDN-1 is associated with the progression and metastasis of colon carcinoma [34,93]. In mouse xenograft studies, tumor growth and metastasis is regulated by genetic modulation of CLDN-1 [94]. The nucleus and cytoplasm of colon carcinoma cells and metastatic lesions showed intensified CLDN-1 expression [34]. Many studies suggest that the genes encoding tight junction proteins (TGPs) in CRC are differentially expressed and involved in the process of invasion and cellular transformation [95]. Several studies reported up-regulation of CLDN-1 in CRC [34,44,96,97]. A similar study showed that CLDN-1 overexpression induced a highly invasive and metastatic potential in CRC cells [34]. Noncancerous cells with normal CLDN-1 expression were found to form a monolayer, whereas cells that overexpressed CLDN-1 grew as aggregates. CLDN-1 regulates cellular morphology and behavior in the colonic epithelium [34,44]. The possible involvement of CLDN-1 in the tumorigenesis of ulcerative colitis (UC)-associated CRC has also been demonstrated [44]. Another study demonstrated that the prognostic factor for CRC is the independent expression of CLDN-1 [98]. Delocalization of CLDN-1 from the membrane to cytoplasm and nuclei of cancer cells supports cancer growth and malignancy [34]. In colon cancer cells, CLDN-1 decreases the expression of E-cadherin by upregulating ZEB-1 repressor resulting in invasion and reduction of anoikis [59].

The level of CLDN-1 mRNA was found to be higher in the distal site of the colon as compared to the proximal site and demonstrated significant effects on xenografted tumors growth in athymic mice by changes in the expression of CLDN-1, showing its role in CRC tumorigenesis [34]. Both mRNA and protein levels of CLDN-1 were found to be upregulated in sporadic human CRC compared to the normal mucosa [34]. Dhawan and colleagues reported that T84 cell transfection with CLDN -1 resulted in aggregation and multilayer formation in transfected T84 cells as compared to the T84 parent cells. The interactions between claudin family members are both homophilic and heterophilic and are considered to play a significant role in the progression of CRC and several other cancers [99]. The progression of colon cancer has been linked with the dysregulation of the CLDN-1 expression causing disorganization of the tight junction fibrils leading to increased paracellular permeability [100]. Increased potential for invasion and metastasis has been demonstrated in xenografts that express CLDN-1 [101]. CLDN-1 positively correlates with CRC cell proliferation and influences the growth and evolution of the tumor. Its expression was also found to be associated with accelerated serrated lesions of CRC and was related to anoikis resistance and cellular dis-cohesion [101]. Moreover, serrated polyps with over-expressed CLDN-1 were found to have a higher potential for the development and progression into higher-grade lesions [101].

Activation of the Wnt signaling pathway is strongly implicated in the development of colorectal cancer [102]. Wnt signaling is activated by the loss of the adenomatous polyposis coli (APC) protein or by the activation of *β*-catenin mutations [103,104]. CLDN-1, one of the target genes in the Wnt signaling pathway, has two *β*-catenin binding sites (TCF/LEF) in its promoter region for the activation of the transcription process [105,106]. A study showed that the expression of CLDN-1 was found to be elevated in the intestinal adenomas of the APC in mice as compared to normal tissue [94]. Additionally, high expressions of CLDN-1 are seen in the dysplastic areas of the colon in patients with chronic inflammatory disease [94]. CLDN-1 was overexpressed in metastatic colorectal cancer (mCRC) samples as compared to normal mucosa with differential expression in other CRC subtypes. Consensus molecular subtype CMS2, transit-amplifying and C5 subtypes of the mCRC exhibited higher expression of CLDN-1 [107].

### 2.4. Claudin-1 and Gastric Cancer 

CLDN-1 is highly expressed in gastric cancers [108,109]. High expression of CLDN-1 was reported in intestinal type gastric cancer that correlated with lymph node metastasis, advanced TNM (classification of malignant tumors) stage, recruitment, and activation of MMP-2 and MMP-9, which are all responsible for enhanced cell invasion and metastasis [60,109]. The invasion of gastric adenocarcinoma cells is associated with the levels of CLDN-1 expression as CLDN-1 is found to be upregulated in gastric carcinoma and participates in the metastatic behavior of these cancer cells [45].

One study demonstrated that the localization and correlation of CLDN-1 expression are linked with anoikis resistance in gastric cancer through mediating membrane *β*-catenin expression and by inducing cell aggregation and inhibiting apoptosis cascade [110]. The authors also observed that the levels of CLDN-1 expression in gastric cancer tissues decreased from well to moderate to poorly differentiated tumors, suggesting that reduced CLDN-1 expression is an adverse prognostic factor predicting a lower survival rate [110]. However, another study showed that in comparison to CLDN-4, the expression of CLDN-1 was higher in well-to-moderately differentiated gastric adenocarcinomas [111].

### 2.5. Claudin-1 and Hypopharyngeal Squamous Cell Carcinoma 

Head and neck squamous cell carcinoma (HNSCC) is the sixth most frequent tumor worldwide [112]. Tissue microarray and immunohistochemistry assays of surgical samples suggested that CLDN-1 expression is increased in squamous cell cancer [46,113]. It has been shown previously that CLDN-1 induces the generation of tumor lymphatic vessels and increases the lymph node metastasis [47]. Additionally, a study demonstrated that CLDN-1 expression in squamous cancers differs in an organ-specific manner [113]. CLDN-1 was found to be upregulated in hypopharyngeal squamous cell carcinoma (HSCC). This study showed a positive association of CLDN-1 expression with the degree of tumor differentiation and lymph node metastasis [61].

### 2.6. Claudin-1 and Hepatocellular Carcinoma

Hepatocellular carcinoma (HCC) is the third major cause of death due to cancer and the fifth most common cancer malignancy worldwide [114]. Studies have reported the involvement of CLDN-1in the tumorigenesis and metastasis of HCC [48,63]. HCC cell line experiments demonstrated the role of CLDN-1 in the process of cancer cell invasion [115]. Primary HCC samples were found to be positive for CLDN-1, suggesting its significant role in the formation of metastasis and hepato-carcinogenesis [115]. Another study demonstrated the involvement of CLDN-1 in the epithelial to mesenchymal transition in HCC and hepato-carcinogenesis [62]. One study established that overexpression of CLDN-1 induces MMP-2 in SNU-354, -423 and -449 HCC cells resulting in increased invasion and migration of the cancer cells compared to the normal liver cells and other CLDN-1 expressing HCC cells such as SNU-398 and SNU-475 [63]. However in another study reduced expression of CLDN-1 was reported to be a marker for a poor prognosis in HCC [49], and a further study showed that reduced expression of CLDN-1 reinforced the invasive and cancer stem cell (CSC) like properties of HCC cell lines (Huh7 and Hep3B) in vitro, while the forced expression of CLDN-1 diminished the CSC-like properties of HCC cells [116].

### 2.7. Claudin-1 and Lung Adenocarcinoma 

Lung cancer is one of the leading causes of death worldwide. Several studies have shown that CLDN-1 has a significant role in the pathogenesis of lung cancer [117,118,119]. In lung cancer, CLDN-1 acts as a cancer invasion/metastasis suppressor [57]. CLDN-1 was found to be associated with increased expressions of cancer metastasis suppressors such as connective tissue growth factor (CTGF), thrombospondin 1 (THBS1), deleted in liver cancer 1 (DLC1), occludin (OCLN), zona occludens 1 (ZO-1) and reduced expressions of cancer metastasis enhancers such as secreted phosphoprotein 1 (SPP1), cut-like homeobox 1 (CUTL1), transforming growth factor-alpha (TGF-*α*), solute carrier family 2 (facilitated glucose transporter) member 3 (SLC2A3) and placental growth factor (PGF) in lung adenocarcinoma [57]. For patients with lung adenocarcinoma, CLDN-1 is a potential drug treatment target and a useful predictor of prognosis. Studies showed that the invasive ability of HOP62 lung adenocarcinoma cells is increased by knockdown of endogenous expression of CLDN-1 [57]. Immunohistochemistry and RT-PCR analysis showed that CLDN-1 is either reduced or undetected in adenocarcinomas [57]. The authors demonstrated that CLDN-1 overexpression inhibited adenocarcinoma cell dissociation in wound-healing time-lapse images [57].

Inflammatory mediators, such as TNF*α*, plays a significant role in the process of tumorigenesis [120]. Studies have shown that CLDN-1 is involved in the mediation of inflammatory responses initiated by TNF*α* in different cancers [64,121,122]. An experiment in human lung cancer cell lines, observed that TNF*α* induced the expression of CLDN-1, and knockdown of CLDN-1 blocked 75% of TNF*α*-induced gene expression. In CL1-5 lung cancer cells, cell migration activity was inhibited by over-expression of CLDN-1 and restored by CLDN-1 knockdown in addition to cell invasion ability. The above findings signify a signal mediator role of CLDN-1 in TNF*α* induced gene expression and cell migration [57]. One study demonstrated that CLDN-1 expression correlated with Ras and epidermal-growth-factor-receptor (EGFR) expression suggesting the involvement of the latter two signaling pathways in the regulation of CLDN-1 in lung adenocarcinoma [119]. The results of this study demonstrated an association between CLDN-1 and Ras/EFGR in the development of lung cancer and the combination of both has strong clinical significance [119]. Surprisingly, as compared to the previous studies that showed that overexpression of CLDN-1 suppressed metastatic abilities of lung adenocarcinoma cells [50,57], this study showed that patients with positive expressions of both CLDN-1 and Ras/EGFR were found to have poor prognosis as compared to CLDN-1(+) Ras/EGFR(−), CLDN-1(−) Ras/EGFR(+), and CLDN-1(−) and Ras/EGFR(−) patients [119]. Another study found that enhanced cell migration by tumor necrosis factor and a similar morphology like fibroblast was found to be reduced by small CLDN-1 interfering RNA in the cells of lung cancer [121]. 

### 2.8. Claudin-1 and Pancreatic Cancer 

Pancreatic cancer is the fourth major cause of deaths caused by cancer worldwide, with a strong capacity for metastasis and recurrence [123]. In pancreatic cancer (PC), increased expression of CLDN-1 was found to be associated with epithelial-mesenchymal transition. CLDN-1 is expressed by ductal pancreatic adenocarcinomas as well as intra-ductal papillary mucinous pancreatic tumors. One study demonstrated that 58% positive CLDN-1 immunostaining in ductal pancreatic adenocarcinomas and intraductal papillary pancreatic tumors [51]. Another study showed that through activation of mitogen-activated protein kinase 2 (MEK2), CLDN-1 was involved in cell dissociation of PC cells [124]. A further study observed the role of CLDN-1 in the progression of human PC using the PANC-1 cell line [64]. Increased expression of PARP [poly-(ADP-ribose) polymerase], an apoptosis marker, and decreased PANC-1 proliferation was observed after treatment with TNF-*α* [64]. Increased proliferation in PANC-1 cells was observed after treatment with TNF-*α* and CLDN-1 siRNA against CLDN-1, suggesting the cells were resistant to TNF-*α*-induced apoptosis when transfected with CLDN-1 siRNA. These findings clearly demonstrated that the CLDN-1 expression plays a role in the proliferation of PC cells [64].

### 2.9. Claudin-1 and Ovarian Cancer 

Ovarian cancer affected a significant number of women worldwide and is the seventh most frequent cause of deaths due to cancer in women [125]. The upregulation of the *CLDN-1* gene is found to be associated with ovarian cancer [52]. Studies have shown that overexpression of CLDN-1 caused reduced cell differentiation and a high invasive growth rate [126]. The role of CLDN-1 has been widely studied in two different types of ovarian cancers, namely, ovarian serous and ovarian endometroid carcinoma [127]. The expression of CLDN-1 was shown to be negatively regulated by microRNA-155 (miR-155) which results in reduced proliferation and invasion of human ovarian cancer-initiating cells [52]. Reports have shown that elevated level of CLDN-11, 4, and 7 promotes the growth of both benign and malignant epithelial ovarian cancers [52]. Extensive studies conducted and analyzed for the association of CLDN-1 with survival and anatomical site showed 85% elevation of CLDN-1 expression [58]. Recently, the level of CLDN-1 expression was also investigated in borderline tumors of the ovary (BOT) [53]. A significantly higher level of CLDN-1 expression was associated with the peritoneal implants and micropapillary patterns that are specifically seen only in serous BOT [53].

### 2.10. Claudin-1 and Oral Squamous Cell Carcinoma 

Almost 90% of all oral carcinomas are oral squamous cell carcinomas (OSCC) [128]. It has been shown previously that the invasive activity of OSCC cells is enhanced by CLDN-1 through activation of MMP-1 and 2, resulting in increased cleavage of Laminin-5 *γ*2 chains. The authors of the study further revealed elevated expression of CLDN-1 in OSC-4 and NOS-2 cell lines which are highly invasive [129]. One study demonstrated the association of high CLDN-1 expression with aggressive histopathologic features such as perineural and vascular invasion and suggested that CLDN-1 might be directly or indirectly involved in the progression of OSCC [130]. Another study found that the absence of CLDN-1 was associated with poorly differentiated tumors [131]. Immunohistochemical analysis revealed that the presence of CLDN-1 in the invasive front of tumor islands was associated with neck mode metastasis. The results obtained from this study further suggested that the expression of CLDN-1 is linked with the recurrence of OSCC [132].

### 2.11. Claudin-1 and Melanoma 

Melanoma, which arises from melanocytes, causes 75% of deaths related to skin cancers [133]. CLDN-1 was found to be upregulated in melanoma tissues [134]. In melanoma, CLDN-1 is abnormally/aberrantly expressed in the cytoplasm of malignant cells and not in the cell membrane. This may be related to its influence on protein kinase C (PKC) activity [38]. PKC activation caused an increase in transcription and protein expression of CLDN-1 and thus, cell motility [38]. When melanoma cells transfected with CLDN-1, it increased the secretions of matrix metalloproteinase- 2 (MMP-2) reflecting its contribution to the cell invasion process. The data from French, et al. supported the hypothesis that the invasive capacity of melanoma cells is increased by cytoplasmic expression of CLDN-1 and not by the elevated nuclear expression of CLDN-1 [56]. In melanoma patients with brain metastases, the expression of CLDN-1 was downregulated, and the introduction of CLDN-1 retrovirus reduced the tumor aggressiveness and tumor migration ability and diminished micro-metastasis in the brain. This shows that reduction in CLDN-1 supports tumor progression and metastasis and that CLDN-1 can be used as a prognostic predictor for melanoma patients with increased risk of brain metastasis [135].

### 2.12. Claudin-1 and Prostate Cancer

Prostate cancer is the second most diagnosed malignancy and fifth-most leading cause of cancer deaths in men [136]. The Gleason grading score system is the most commonly used method to evaluate the aggressiveness of prostate cancers, whereas, the changes in the glandular architecture indicate the tumor grades [54]. The typical glandular architecture is supported by cellular polarity and cell-to-cell contact, and thus the alterations and dysregulations of proteins mediating normal cellular connection may impact the histology and Gleason grade [54]. It is suggested that the loss of tight junction protein, CLDN-1, is associated with cancer invasion, progression and the transformation into metastatic phenotype in prostate cancers. A study reported that the lower expression of CLDN-1 correlated with higher prostate-specific antigen in prostate cancer [54]. Further studies are needed to thoroughly investigate the association between prognostic factors and claudins expression in prostate cancer.

## 3. Claudin-1 and Tight Junction Barrier Function

CLDN-1 is an integral membrane protein that in conjunction with other claudins forms the tight junctions and together plays an essential role in epithelial barrier functions. CLDN-1 has a significant role in epithelial differentiation and loss of CLDN-1 can impair the functioning of tight junctions [137]. Numerous studies have reported the involvement of CLDN-1 in transepithelial electrical resistance (TER) [138,139] and paracellular permeability [140] showing its importance in tight junction barrier functions. The study showed that CLDN-1 increased the TER and reduced paracellular flux in Madin-Darby Canine Kidney (MCDK) cells [138]. While another study reported that induced CLDN-1-myc in MDCK cells resulted in the formation of aberrant tight junction strands independently without the participation of ZO-1 and occludin [139]. CLDN-1 controls the flux of solutes by localizing at tight junctions and modulating the paracellular permeability. Any defect in the expression of CLDN-1 can result in tight junction dysfunction causing increased paracellular permeability leading to various pathologies such as in Neonatal ichthyosis-sclerosing cholangitis (NISCH) syndrome [140]. This study showed that the silencing of CLDN-1 leads to increased hepatic paracellular permeability [140]. Mostly, an increase in the tight junction proteins should lead to an increase in the tight junction integrity but this is not a universal phenomenon as one study reported that high expression of CLDN-1 resulted in decreased TER and increased permeability causing loss of barrier function in intestinal epithelial cells (IEC-18) treated with TNF-*α* [141]. The authors explained the reason for this contradictory finding is that the loss of barrier function was due to the reduced expression of occludin protein and not CLDN-1 [141]. Besides, the other study reported that high expression of CLDN-1 resulted in blood–brain-barrier (BBB) leakiness during post-stroke recovery and targeting of CLDN-1 by a CLDN-1 peptide improved the permeability of brain endothelial barrier [142]. So, these studies suggest that the upregulation of CLDN-1 cannot necessarily be universal to the increased barrier function and there might be other contributing factors that regulate these functions, and also we should not rule out the tissue specific expression of claudins as other possibility to their dichotomous roles.

## 4. Gene Expression, Survival and Pathway Interaction Analysis of claudin-1 across different Cancers

Survival Analysis was performed using online survival analysis tools to assess the influence of CLDN-1 expression on survival in different types of cancer using the Cancer Genome Atlas (TCGA) datasets. This analysis was to investigate whether an alteration in gene expression correlates with poor survival or with tumor recurrence. The data showed that the gene expression of CLDN-1 did not affect survival significantly in cancers like breast invasive carcinoma (BRCA), cervical squamous cell carcinoma and endocervical adenocarcinoma (CESC) and pancreatic adenocarcinoma (PAAD) (*p* > 0.05), but did significantly associate with survival for cancers like thyroid carcinoma (THCA), adrenocortical carcinoma (ACC), rectum adenocarcinoma (READ) (* *p* < 0.05) (Figure 4). All survival analysis was performed using online survival analysis tools.

We further compared the expression of CLDN-1 between tumor and normal tissues for ACC, CESC, PAAD, READ, THCA and BRCA from TCGA datasets. We found that most of the cancers showed significant expression differences of CLDN-1 between tumor and normal type. We observed that for most of the cancers, tumor tissues have higher expression than normal tissues but for cancer like ACC, CLDN-1 expression in normal tissue was found to be greater than tumor tissues (Figure 5). We did not find any significant difference in CLDN-1 expression between tumor and normal tissue of BRCA.

To determine the interaction of CLDN-1 with other genes, we performed gene interaction analysis using the Gene MANIA prediction server. We found that CLDN-1 significantly interacted with several key genes that play an important role in normal cell physiology. Any disturbance in CLDN-1 expression or its partners may result in the manifestation of various diseases including cancers. The interaction of CLDN-1 with other key molecules can be individually analyzed in different cancers. This may highlight the key pathways which can be therapeutically targeted to suppress cancer growth or metastasis (Figure 6).

## 5. Claudin-1 as a Drug Target

The involvement of CLDN-1 in various pathological conditions has provided new perceptions into drug development targeting CLDN-1. The approach of targeting CLDN-1 either by monoclonal antibodies or chimeric antibodies has great potential but needs more research to reach the level of clinical trials. The initial studies have laid an important foundation towards the new strategies that could be employed and further modified towards the potential usefulness of CLDN-1 as a therapeutic target. The localization of CLDN-1 as a transmembrane protein makes it a perfect target for the enhanced drug absorption for preventing infection and treating cancer. One of the studies observed that the human hepatocytes treated with mouse anti-CLDN-1 monoclonal antibodies (mAbs), showed improved drug absorption and prevented hepatitis C virus (HCV) infection [143]. A human-mouse chimeric CLDN-1 mAb (clone 3A2) demonstrated cellular cytotoxicity against CLDN-1 expressing cancer cells [144]. The other aspect of claudins that is being exploited for therapeutic targeting is their role in regulating paracellular permeability in different tissues. *Clostridium perfringens* enterotoxin (cCPE) binds with claudin through its claudin binding domain and inhibits the claudin function. It was observed that blocking CLDN-1 with cCPE variants in the Huh7.5 hepatoma cell line inhibited infection of Huh7.5 cells with HCV in a dose-dependent manner and this also opened the epidermal barrier in the reconstructed human epidermis [145]. To eliminate the possible limitation facing CLDN-1 targeted therapies due to genotype-dependent escape via CLDN-6 and CLDN-9 and to improve anti-HCV activity, humanized anti- CLDN-1 monoclonal antibody (mAb) could be an alternative. One study developed functional mAB against extracellular domains of CLDN-1 and found that these antibodies have a very high affinity for intact CLDN-1, efficiently inhibited HCV infections both in vitro and in vivo, further demonstrating that anti-CLDN1 mAbs could be useful in inhibiting HCV infections [146]. In a very recent study, CLDN-1 was successfully targeted with anti-CLDN1 near-infrared fluorophore to track the colorectal cancer cells, and it may provide a novel way for fluorescence-guided surgery of tumor [147].

The main concern with claudin-targeted therapies is the presence of claudins in both normal epithelial cells and cancer cells that makes the targeting difficult. However, it has been observed that claudins are localized at the tight junctions in normal tissues, while in malignant tissues, there is a dysregulation of claudins localization from the tight junctions to the cell surface [24,148]. Claudins with aberrant localization in malignant tumors can be recognized by utilizing the C-terminal claudin-binding domain of cCPE fused with protein synthesis inhibitory factor (C-CPE-PSIF), causing less cytotoxicity to normal cells and a study has shown how CLDN-4 can be used as a target for tumor therapy by fusion of cCPE with PSIF. The results of the study showed that C-CPE-PSIF was cytotoxic to cells with undeveloped tight junctions (preconfluent cultures of Caco-2) and was not cytotoxic to cells with developed tight junctions (postconfluent cultures of Caco-2) [149]. Several other studies have used CLDN-4 as a target for tumor therapy and showed the accumulation of anti-CLDN-4 mAbs specifically in the tumors and reduced the growth of human colorectal and gastric tumors in mice [150]. One study detected CLDN-4 upregulation non-invasively in mice pancreatic ductal adenocarcinoma xenografts by using MRI and ^18^FDG-PET [151] (for detailed reviews see [152]). In the future, a similar approach can be applied to CLDN-1 by preparation of CLDN-1 targeting molecule and can be tested for cytotoxicity to normal cells.

## 6. Claudins and Autosomal Recessive Disorders

As explained in the previous sections, the claudin family of proteins is an integral part of tight junctions that determine paracellular selectivity and permeability to small ions by acting as pores or barriers in polarized epithelia. We also discussed how overexpression or reduction of claudins could both promote and limit cancer progression, revealing complex dichotomous roles for claudins depending on cellular context. Besides the fact that the abnormal or deregulated expression of claudins has been associated with different human diseases like cancer, there are also other human disorders such as autosomal recessive disorders that have been reported due to clearly defined mutations in the corresponding claudin genes. Such disorders are mostly observed in skin, liver, kidney, the inner ear, and the eye. The first evidence that showed a mutation in the claudin family of tight junction proteins causes human disorders were from the group of Lifton [153]. In their study, they reported that the mutations in the human gene, paracellin-1 (PCLN-1)/CLDN-16 causes an autosomal recessive disorder called Familial hypomagnesemia with hypercalcinuria and nephrocalcinosis (FHHNC) characterized with renal Mg^2+^ and Ca^+^ wasting. Later the same group revealed additional evidence that loss of function mutations in paracellin-1 PCLN-1/CLDN-16, are causative of FHHNC [154]. PCLN-1 is related to the claudin family of tight junction proteins and is in tight junctions of the thick ascending limb of Henle (TAL). CLDN-16 is a cattle ortholog of PCLN-1 with ∼ 90% sequence homology, and PCLN-1/CLDN16 mutations have been shown to be strongly associated with bovine chronic interstitial nephritis with diffuse zonal fibrosis (CINF) [155]. Although both renal disorders FHHN and CINF are caused by PCLN-1/CLDN16 mutations, but the clinical features of both diseases are quite different which may be due to specific mutations/deletions in the same gene or through species specificity. Since the first report [153], several other tight junction disorders have been shown to cause human diseases including mutations in claudin proteins such as CLDN-1 [156], CLDN-9 [157], CLDN-10 [158], CLDN-14 [159], CLDN-16 [160], CLDN-19 [161] (for detailed reviews see [162]).

## 7. Conclusions

It is clear from the literature that among all the tight junction proteins, the claudin family of proteins is particularly important in regulating normal cell physiology. Among the claudin family, CLDN-1 is the most extensively studied protein and has been shown to be involved directly or indirectly in the development and progression of cancer, and also has a suppressive role in some cancers. CLDN-1 acts alone or in combination with other molecules to exert its tumor promoting or suppressing effect. Likewise, the shuttling of CLDN-1 between the cell membrane, cytoplasm and nucleus is a deciding factor in the development and progression of cancers. Another important aspect is the involvement of CLDN-1 in many signaling pathways, especially in Wnt and Notch signaling. The association of CLDN-1 with patient survival or recurrence in many cancers suggests its importance as a prognostic marker and as a potential therapeutic target. Also, pathway interaction analysis revealed CLDN-1 interacting partners, which can be further explored as potential drug targets. Based on the complexity of the topic, there is no one statement we can make for CLDN-1 role in cancer or barrier function since it is more intricate (Claudins are upregulated or downregulated in cancer and may or may not play a role in barrier function). In other words, universal statements concerning CLDN-1 and cancer or CLDN-1 and barrier function are dangerous oversimplifications.

## Figures and Tables

**Figure 1 ijms-21-00569-f001:**
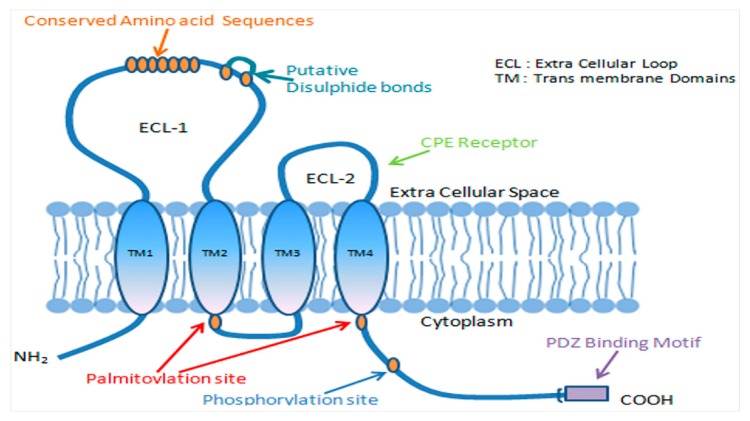
A schematic presentation of tight junction complex involving claudins and other major components. Claudins contain four transmembrane domains (TMD-1, TMD-2, TMD-3, and TMD-4) and two extracellular (ECL) loops. The PDZ-binding domain of the –COOH terminal of claudin undergoes post-transcriptional modification and has been implicated in signal transduction.

**Figure 2 ijms-21-00569-f002:**
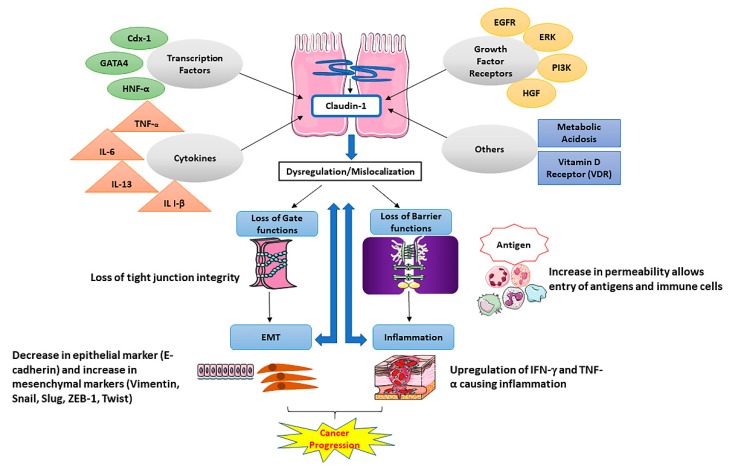
Schematic model of role and regulation of CLDN-1 in a normal or diseased state. In normal physiological conditions, CLDN-1 expression/ integrity is regulated by transcription factors, growth factors and cytokines, which in turn maintain the normal gate function and barrier function of tight junctions. Dysregulation of CLDN-1 expression can result in the compromise of membrane barrier functions and gate functions, which subsequently can lead to the upregulation of the expression of pro-inflammatory markers such as IFN-*γ* and TNF-*α*. In cancer, the loss of CLDN-1 facilitates the malignant transformation of cancer cells and epithelial-mesenchymal transition (EMT).

**Figure 3 ijms-21-00569-f003:**
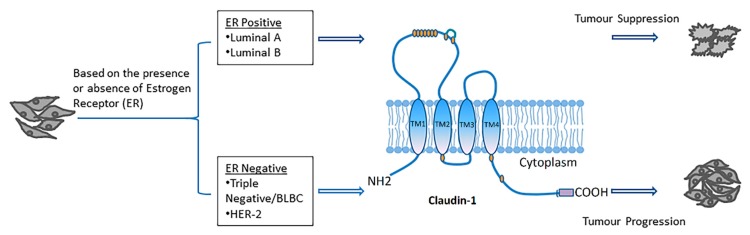
CLDN-1 expression in different subtypes of breast cancer as characterized by the presence or absence of estrogen receptor (ER). Luminal A, and Luminal B subtypes of human invasive breast cancer (ER-positive) exhibit low levels of CLDN-1, which suggests the suppressor role of CLDN-1 in these tumors. However, aggressive forms (ER-negative) exhibit overall high levels of CLDN-1 expression, which signifies CLDN-1 role as a tumor promoter.

**Figure 4 ijms-21-00569-f004:**
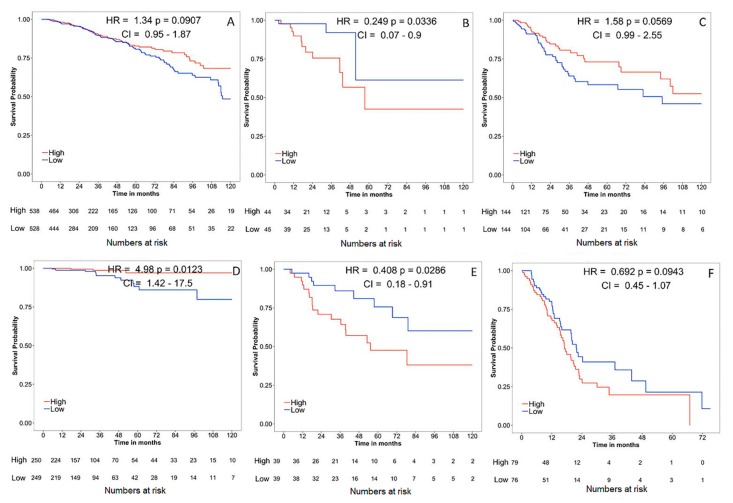
Survival analysis of CLDN-1 in various cancers. The red line denotes higher expression, and the blue line indicates lower expression. (**A**) Breast invasive carcinoma (BRCA); (**B**) rectum adenocarcinoma (READ); (**C**) cervical squamous cell carcinoma and endocervical adenocarcinoma (CESC); (**D**) thyroid carcinoma (THCA); (**E**) adrenocortical carcinoma (ACC); (**F**) pancreatic adenocarcinoma (PAAD).

**Figure 5 ijms-21-00569-f005:**
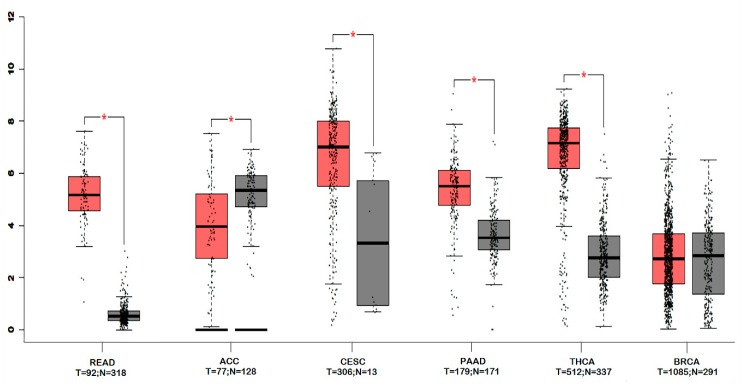
Boxplot showing the distribution of CLDN-1 expression in tumors and normal tissues for different types of cancers. Significant differences are shown with an asterisk (*). Boxplots were generated using GEPIA1 webserver and *p*-value < 0.01 was considered as significant.

**Figure 6 ijms-21-00569-f006:**
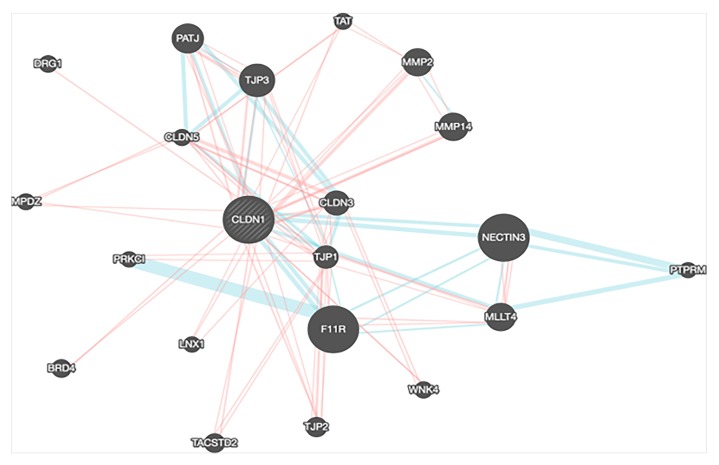
CLDN-1 interaction network using the Gene MANIA prediction server.

**Table 1 ijms-21-00569-t001:** Expression of CLDN-1 in different types of cancer.

Type of Cancer	Subtypes	Expression of CLDN-1	References
Breast Cancer	Luminal A	Downregulated	[40]
	Luminal B	Downregulated	[40]
	Triple negative/Basal like	Upregulated	[40]
	HER2 enriched	Downregulated	[41]
	Claudin-low	Downregulated	[41]
Thyroid Cancer	Papillary Thyroid Cancer	Upregulated	[42]
	Follicular Thyroid Cancer	Upregulated	[43]
Colorectal Cancer	Ulcerative Colitis associated Colorectal Cancer	Upregulated	[44]
	Sporadic Colorectal Cancer	Upregulated	[34]
Gastric Adenocarcinoma	-	Upregulated	[45]
Head and Neck Squamous Cell Carcinoma	-	Upregulated	[46]
Hypopharyngeal Squamous Cell Carcinoma	-	Upregulated	[47]
Hepatocellular Carcinoma	-	Downregulated	[48,49]
Lung Adenocarcinoma	-	Downregulated	[50]
Pancreatic Ductal Carcinoma	-	Upregulated	[51]
Epithelial Ovarian Cancer	-	Upregulated	[52,53]
Oral Squamous Cell Carcinoma	-	Upregulated	[35]
Melanoma	-	Upregulated	[38]
Prostate adenocarcinoma	-	Downregulated	[54,55]

“-“ no subtypes.

**Table 2 ijms-21-00569-t002:** Role of CLDN-1 in different cancers.

Cancer Type	Activity	Findings	References
**Melanoma**	Tumor Promoter	Cytoplasmic expression of CLDN-1 contributes to the migratory capacity of melanoma cells	[56]
**Oral Squamous Cell Carcinoma**	Tumor Promoter	CLDN-1 enhances the invasive activity of OSC-4 and NOS-2 cell lines by activation of MT1-MMP and MMP-2	[35]
**Prostate Cancer**	Tumor Suppressor	Loss of CLDN-1 associated with progression of Prostate cancer	[54]
**Lung Cancer**	Tumor Suppressor	Knockdown of CLDN-1 increased invasive and metastatic activity of lung adenocarcinoma cells	[57]
**Breast Cancer**	Tumor Promoter in ER-Subtypes	Increases cell migration and also exhibits an anti-apoptotic effect	[40]
	Tumor Suppressor in ER+ Subtypes	Acts as a suppressor of mammary epithelial proliferation Increases apoptosis of breast cancer cells	
**Thyroid Cancer**	Tumor Promoter	High expression of CLDN-1 found in follicular thyroid carcinoma (FTC-133) and Papillary Thyroid Carcinoma cells	[42,43]
**Ovarian Cancer**	Tumor Promoter	High expression of CLDN-1 correlated with shorter overall survival in ovarian carcinoma effusions	[58]
**Colon Cancer**	Tumor Promoter	High CLDN-1 expression in colon carcinoma and metastasis CLDN-1 upregulates the repressor ZEB-1 to reduce expression of E-cadherin in colon cancer cells	[34,59]
**Gastric Cancer**	Tumor Promoter	High expression of CLDN-1 in gastric cancer associated with poor survival	[60]
**Hypopharyngeal Squamous Cell Carcinoma**	Tumor Promoter	High expression of CLDN-1 associated with lymph node metastasis and degree of tumor differentiation	[61]
**Hepatocellular Carcinoma**	Tumor Promoter	CLDN-1 promoted epithelial-mesenchymal transition (EMT) in HCC cells by overexpression of mesenchymal markers (N-cadherin and vimentin)	[48,62,63]
**Pancreatic Cancer**	Tumor Promoter	TNF-α upregulated CLDN-1 expression, leading to increased proliferation of pancreatic cancer cells	[64]

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
