# Peer review of "Claudin-1, A Double-Edged Sword in Cancer"

_ijms, 2020, doi:10.3390/ijms21020569_

Round 1

Reviewer 1 Report

A review of any claudin and cancer is really a herculean task.  The authors do a scholarly job of trying to cover what is in the published literature.  It is a thankless task because of the maddening complexity and even contradictoriness of their subject matter.  

I detected a lot of grammar issues.  I've made corrections where I saw them.  I'm enclosing a file where changes are made in Tracking Mode.  I hope its helpful.  You should scour the ms for more.  I also include suggested changes where the meaning could be obscured.

The biggest problem with this topic is that there seems to be no universal here.  For example I don't see a subclass of cancers where CL-1 is ALWAYS upregulated or ALWAYS downregulated.  There seem to be a myriad of "exceptions."  If there ARE such universals - like a subclass of breast cancers where e.g. CL-1 is ALWAYS downregulated then I strongly recommend you make a summary table or a concluding bulletted paragraph where you show exactly that.  

In general, here are specific issues that I have:

You often conflate tight junctions with "cell adhesion."  Cell adhesion classically is the domain of cadherins and integrins.  Tight junctions - and thus claudins - are a distinct class.  (see e.g. line 90).    

There are many statements made throughout the review that CL-1 is CAUSING certain changes/effects.  Some, though perhaps not all, should be changed to "associations" and "correlations."  I'd suggest the authors scour the review for such causal statements and see if it really is a causal effect - in each instance.  If it IS a true causal effect, then I would describe in an additional sentence why it is causal, and not merely associative.  See e.g. Table 1 where you say CL-1 PROMOTES cell invasion.  So I ask, IS it really causing this or associating with this?  There are many examples throughout the text as well. e.g. lines 200-201.    

There is an intriguing question left unaddressed in your review.  I'd deal with it.  Discuss whether CL-1 is "good" for tight junction barrier function.  Is more CL-1 (upregulation) good for barrier function, i.e. a tighter junction?  Or not? Or one can't provide a universal, yes or no?  This is an important parallel topic to your cancer main topic.  I think you could do it in one page or less.  

Similarly, I'd have a section dealing with the issue of whether cancers universally have failed tight junctions, i.e. loss of barrier function.  I don't think that this is true.

Author Response

Dear reviewers and editor,

Thank you for the opportunity to revise our review paper entitled “Claudin-1; A Double-Edged Sword in cancer”.

We have included the reviewer reports in this letter and included a point-by-point response describing the changes we have made. The changes are marked as track changes in the paper, as you requested. 

Comments and Suggestions for Authors

Reviewer 1

A review of any claudin and cancer is really a herculean task.  The authors do a scholarly job of trying to cover what is in the published literature.  It is a thankless task because of the maddening complexity and even contradictoriness of their subject matter.  

The biggest problem with this topic is that there seems to be no universal here.  For example, I don't see a subclass of cancers where CL-1 is ALWAYS upregulated or ALWAYS downregulated.  There seem to be a myriad of "exceptions."  If there ARE such universals - like a subclass of breast cancers where e.g. CL-1 is ALWAYS downregulated, then I strongly recommend you make a summary table or a concluding bulletted paragraph where you show exactly that.  

Response: We thank the reviewer for careful reading and pointing out scientific errors in our manuscript. We apologize for the unintentional mistakes in the paper which are fixed in the revised manuscript. As per reviewer suggestions, we have included a new table (Table 2) in the revised manuscript which clearly depicts the role played by CLDN-1 in different cancers and cancer subtypes. In addition, we also made few changes in Table 1 to portray the conclusive findings of CLDN-1 in different cancers. The general picture about the role of claudins is that they function in tissue specific and context dependent manner. In most of the cancers, CLDN-1 either associates or correlates with any changes or affects such as tumor cell invasion or proliferation but in certain cancer types, reports reveal the highly contextual upregulation of claudin expression and their relevance in promoting cancer cell invasion and metastatic progression. For example, in human liver cells, CLDN-1 was found to promote epithelial-to-mesenchymal transition via the c-Abl/Raf/Ras/ERK signaling pathway (Y Suh et al. Oncogene 32 (41), 4873-82. 2013)   and has a causal role in the acquisition of invasive capacity through c-Abl-protein Kinase Cdelta (PKCdelta) signaling (CH Yoon et al. J Biol Chem 285 (1), 226-33. 2010).

You often conflate tight junctions with "cell adhesion."  Cell adhesion classically is the domain of cadherins and integrins.  Tight junctions - and thus claudins - are a distinct class.  (see e.g. line 90).  

Response:   We thank the reviewer for raising this question and we do agree with the statement, though there are studies which have linked claudins and cell adhesion molecules (The Structure and Function of Claudins, Cell Adhesion Molecules at Tight Junctions. S Tsukita, M Furuse, Ann N Y Acad Sci, 915, 129-35, 2000). We tried to keep them as separate entities in our revised manuscript.

There is an intriguing question left unaddressed in your review.  I'd deal with it.  Discuss whether CL-1 is "good" for tight junction barrier function.  Is more CL-1 (upregulation) good for barrier function, i.e. a tighter junction?  Or not? Or one can't provide a universal, yes or no?  This is an important parallel topic to your cancer main topic.  I think you could do it in one page or less.  

Response: We thank the reviewer for insightful suggestions. We have included a separate section (Caludin-1 and tight junction barrier function) in the revised manuscript and tries to be detailed information to the best of our knowledge.

Similarly, I'd have a section dealing with the issue of whether cancers universally have failed tight junctions, i.e. loss of barrier function.  I don't think that this is true

Response: We thank the reviewer for valuable thoughts which helped us in improving the current version of the article. We have included an additional section under the heading “Tight junctions” in our revised manuscript which deals with the issue in a fair manner.

Reviewer 2 Report

This is a well written manuscript.

Do the authors think that autosomal ressecive claudins related diseases are prone to cancer ? This point may be briefly mentioned in the discussion.

Author Response

Dear reviewers and editor,

Thank you for the opportunity to revise our review paper entitled “Claudin-1; A Double-Edged Sword in cancer”.

We have included the reviewer reports in this letter and included a point-by-point response describing the changes we have made. The changes are marked as track changes in the paper, as you requested. 

Reviewer 2

Comments and Suggestions for Authors

This is a well written manuscript.

Do the authors think that autosomal recessive claudins related diseases are prone to cancer? This point may be briefly mentioned in the discussion.

Response: We highly appreciate the reviewer for raising an insightful question. As per the reviewer’s suggestion, we have included a separate section (Claudins and Autosomal Recessive Disorders) in our revised manuscript.

Reviewer 3 Report

The paper is well written and can be accepted for publication.

I think the manuscript is very well-written. It is an excellent summary of the field and covers all the relevant science. I feel the paper is essentially ready to be published.

Author Response

Dear reviewers and editor,

Thank you for the opportunity to revise our review paper entitled “Claudin-1; A Double-Edged Sword in cancer”.

We have included the reviewer reports in this letter and included a point-by-point response describing the changes we have made. The changes are marked as track changes in the paper, as you requested. 

Reviewer 3

Comments and Suggestions for Authors

The paper is well written and can be accepted for publication.

Response: We highly appreciate the reviewer for liking the concept that our manuscript displays.

Round 2

Reviewer 1 Report

The authors were wise to include Table 2.  It undercuts a (mis)conception that cancer involves less expression of Claudin-1 but it is intellectually honest and reflects the complexity of this topic.

I would add in your conclusion a statement that universal statements concerning claudin-1 and cancer or claudin-1 and barrier function are dangerous oversimplifications.

Author Response

Dear reviewers and editor,

Thank you for the opportunity to revise our review paper entitled “Claudin-1; A Double-Edged Sword in cancer”.

We have included the reviewer reports in this letter and included a point-by-point response describing the changes we have made. The changes are marked as track changes in the paper, as you requested. 

Reviewer 1

Comments and Suggestions for Authors

The authors were wise to include Table 2.  It undercuts a (mis)conception that cancer involves less expression of Claudin-1 but it is intellectually honest and reflects the complexity of this topic.

Response: We highly appreciate the reviewer for liking the table in revised manuscript and we agree with the views of reviewer about the complexity of the topic.

I would add in your conclusion a statement that universal statements concerning claudin-1 and cancer or claudin-1 and barrier function are dangerous oversimplifications.

Response: We agree with the reviewer’s statement and have included in the conclusion of the revised version.
